# Robust Localization of Industrial Park UGV and Prior Map Maintenance

**DOI:** 10.3390/s23156987

**Published:** 2023-08-06

**Authors:** Fanrui Luo, Zhenyu Liu, Fengshan Zou, Mingmin Liu, Yang Cheng, Xiaoyu Li

**Affiliations:** 1School of Information Science and Engineering, Shenyang University of Technology, Shenyang 110870, China; 202120679@smail.sut.edu.cn (F.L.); tanxiaojian@smail.sut.edu.cn (Y.C.); lxy_4210@smail.sut.edu.cn (X.L.); 2SIASUN Robot & Automation Co., Ltd., Shenyang 110169, China; zoufengshan@siasun.com (F.Z.); liumingmin@siasun.com (M.L.)

**Keywords:** LiDAR-IMU SLAM, feature extraction, prior map maintenance, relocation, unmanned ground vehicle

## Abstract

The precise localization of unmanned ground vehicles (UGVs) in industrial parks without prior GPS measurements presents a significant challenge. Simultaneous localization and mapping (SLAM) techniques can address this challenge by capturing environmental features, using sensors for real-time UGV localization. In order to increase the real-time localization accuracy and efficiency of UGVs, and to improve the robustness of UGVs’ odometry within industrial parks—thereby addressing issues related to UGVs’ motion control discontinuity and odometry drift—this paper proposes a tightly coupled LiDAR-IMU odometry method based on FAST-LIO2, integrating ground constraints and a novel feature extraction method. Additionally, a novel maintenance method of prior maps is proposed. The front-end module acquires the prior pose of the UGV by combining the detection and correction of relocation with point cloud registration. Then, the proposed maintenance method of prior maps is used to hierarchically and partitionally segregate and perform the real-time maintenance of the prior maps. At the back-end, real-time localization is achieved by the proposed tightly coupled LiDAR-IMU odometry that incorporates ground constraints. Furthermore, a feature extraction method based on the bidirectional-projection plane slope difference filter is proposed, enabling efficient and accurate point cloud feature extraction for edge, planar and ground points. Finally, the proposed method is evaluated, using self-collected datasets from industrial parks and the KITTI dataset. Our experimental results demonstrate that, compared to FAST-LIO2 and FAST-LIO2 with the curvature feature extraction method, the proposed method improved the odometry accuracy by 30.19% and 48.24% on the KITTI dataset. The efficiency of odometry was improved by 56.72% and 40.06%. When leveraging prior maps, the UGV achieved centimeter-level localization accuracy. The localization accuracy of the proposed method was improved by 46.367% compared to FAST-LIO2 on self-collected datasets, and the located efficiency was improved by 32.33%. The *z*-axis-located accuracy of the proposed method reached millimeter-level accuracy. The proposed prior map maintenance method reduced RAM usage by 64% compared to traditional methods.

## 1. Introduction

With the rapid development of science and technology, the construction of intelligent and digital industrial parks has entered a transformative phase [1,2]. Ensuring the stable operation of UGVs within industrial parks and guaranteeing their accurate arrival at designated positions are key challenges that must be addressed. The accuracy and efficiency of pose estimation serve as fundamental prerequisites for achieving the reliable arrival of UGVs at the designated position [3,4,5]. Simultaneous localization and mapping (SLAM) techniques can use external information obtained by sensors to perform real-time 6DOF state estimation, effectively resolving the problem of precise localization in the absence of prior GPS measurements, thus providing a crucial foundation for UGV operations within industrial parks [6]. SLAM technology can be classified into visual and LiDAR SLAM. Visual SLAM, which relies on camera-based scene feature extraction, is susceptible to lighting conditions and environmental textures, making it unsuitable for real-time UGV localization [7]. On the other hand, LiDAR SLAM compares the emission signals of laser beams with the received echo signals to obtain point cloud information from the scene, which is less affected by environmental factors [8]. Therefore, LiDAR SLAM is better suited to real-time UGV localization within industrial parks.

### 1.1. Odometry Based on LiDAR-IMU

Currently, LiDAR SLAM can be classified into loosely coupled and tightly coupled methods, based on the approaches used for LiDAR-IMU fusion [9]. Loosely coupled methods involve the direct integration of IMU pre-integration results into the front-end of the LiDAR odometry. These pre-integration results are utilized as initial estimates for LiDAR scanning registration, and they serve as prior values for distortion compensation. The IMU pre-integration results are not incorporated into the joint optimization process. Zhang et al. [10] proposed LiDAR odometry and mapping (LOAM), which is a widely recognized and classic loosely coupled method for fusing LiDAR and IMU data. LOAM has been extensively evaluated on the KITTI dataset [11], demonstrating outstanding performance, and serving as a milestone in the field. However, ensuring its robustness in complex environments is challenging, due to its heavy reliance on stable edge and planar features present in the environment. Shan et al. [12] proposed a lightweight and ground-optimized version of LOAM (LeGO-LOAM), based on the original LOAM. LeGO-LOAM adopts a two-step L-M optimization approach on separated ground points and on points representing edges and planes in space. This method conducts separate three-degrees-of-freedom pose estimations and combines them to achieve a robust six-degrees-of-freedom pose estimation. The aforementioned method can enhance pose estimation accuracy while concurrently reducing system computational complexity. However, its heavy reliance on ground features poses challenges to achieving robust pose estimation in environments characterized by complex ground features. On this basis, reference [13] proposed Livox LOAM specifically designed for solid-state LiDAR systems, and reference [14] proposed M-LOAM for simultaneous localization and mapping utilizing multiple LiDARs. These methods are classified as loosely coupled LiDAR-IMU methods.

Unlike loosely coupled methods, tightly coupled methods typically incorporate the original feature points from LiDAR into IMU data, and they integrate the IMU pre-integration results into joint optimization. Tightly coupled methods offer the advantage of mitigating drift by incorporating residual calculations of point-to-edge and point-to-plane distances, especially in scenarios where the quality of point cloud scene features deteriorates. Tightly coupled methods can be primarily categorized into optimization-based methods and filtering-based methods. Ye et al. [15] proposed a tightly coupled LiDAR-IMU method known as tightly coupled 3D LiDAR industrial odometry and mapping (LIOM). The LIOM method mitigates IMU and LiDAR errors by optimizing the sliding window and incorporating rotation-constrained refinement. It achieves reliable pose estimation in scenarios with reduced feature availability, via the tightly coupled approach. However, this method suffers from high computational complexity and significant time consumption. Qin et al. [16] proposed a lightweight tightly coupled method known as robocentric LiDAR industrial state estimator for robust and efficient navigation (LINS). This method employs an iterated-error-state Kalman filter (ESKF) to achieve the tight fusion of LiDAR and IMU measurements, resulting in enhanced efficiency without compromising accuracy. Shan et al. [17] proposed a tightly coupled LiDAR inertial odometry via the smoothing and mapping (LIO-SAM) method, based on factor graph optimization. This method introduces a sliding window of key frames from the LiDAR data, effectively reducing the computational complexity. The factor graph incorporates IMU pre-integration factors, LiDAR odometry factors, loop closure factors, and GPS factors for joint optimization, leading to improved precision. Xu et al. [18,19] proposed FAST-LIO and FAST-LIO2, which utilize the iterative extended Kalman filter (IEKF) for tightly coupled LiDAR-IMU fusion. FAST-LIO introduces a novel Kalman gain calculation formula inspired by the matrix inverse lemma. This formula transforms the dimension of the inverse matrix from the measurement dimension to the state dimension. Motion distortion is compensated for using a backward propagation method. FAST-LIO2 significantly reduces computational burden and enables high-frequency odometry output at 50 Hz. To enhance computational efficiency, FAST-LIO2 introduces an incremental k-d tree for map maintenance, leading to an accelerated K-nearest neighbor (KNN) search speed and improved computational efficiency.

### 1.2. The Problems in the Real-Time Localization of UGV in Industrial Parks

To fulfill the demands of high-precision and high-frequency odometry for path planning and motion control of UGVs within industrial parks, LiDAR SLAM encounters the following novel challenges:Given the intricate and expansive nature of industrial parks, traditional algorithms encounter challenges in mitigating odometry drift in extensive environments and over long distances. The utilization of low-frequency odometry with inadequate real-time performance can result in errors and delays in UGV motion control. Consequently, traditional methods struggle to fulfill the high-frequency and real-time localization demands of UGVs [20,21,22,23].By leveraging prior maps for real-time localization, the localization accuracy of UGVs within industrial parks can be significantly enhanced. However, the utilization of prior maps for large-scale industrial parks presents challenges, as it necessitates substantial RAM occupancy. Consequently, performing a KNN search on UGVs with limited RAM capacity becomes arduous and impractical [24].The edge and planar features in industrial park environments exhibit relative prominence; however, they are often accompanied by numerous unstable features. Traditional methods encounter challenges in efficiently and accurately extracting high-quality feature points, as well as in concurrently extracting ground points in a targeted manner [25,26,27,28].

Reference [29] proposed utilizing prior maps to compute the residuals of feature point clouds, enabling the generation of highly precise and high-frequency poses that fulfill the odometry accuracy demands of UGVs. Many existing methods maintain a prior map using Octree or kd-tree structures. While the speed of KNN search may satisfy the requirements, the associated RAM usage should not be overlooked, especially when dealing with sizable prior maps. The limited RAM capacity of UGV host systems often poses a challenge in meeting these requirements. References [30,31,32] employed calculations of structural tensors, point feature histograms, and viewpoint feature histograms to extract feature points. While these methods demonstrate acceptable performance in feature extraction, their extraction speed is inadequate to meet the requirements of high-frequency odometry. The curvature feature extraction method categorizes feature points into planar points and edge points. However, this method exhibits poor performance in accurately extracting stable ground points as the characteristics of ground points on LiDAR scanning lines differ from planar points. Hence, further improvements are necessary to enhance the speed and accuracy.

### 1.3. New Contributions

In summary, to fulfill the real-time localization requirements of UGVs in industrial parks, this study presents a tightly coupled LiDAR-IMU odometry method that leverages prior maps. The key contributions of this research can be outlined as follows:A method is proposed for extracting high-quality feature points using a bidirectional projection plane slope difference filter. This method not only reduces the feature extraction speed but also enhances the quality of feature points. Moreover, it enables the separate extraction of ground points and planar points.A maintenance method is proposed for large prior maps, which integrate three-layer voxels with an ikd-tree structure. This method effectively reduces RAM consumption and enables the real-time localization of UGVs using the large prior maps.A back-end optimization model incorporating ground constraints is proposed. This method assigns adaptive weights to observation error equations of ground and planar points using the proposed pseudo occupancy method. This method addresses the issue of inadequate lateral or longitudinal constraints that may arise due to a limited number of points on the plane or ground.

## 2. Overall System Framework

This paper presents a real-time localization method for UGVs using prior maps. The method involves searching for the nearest neighbor points in the observation error equation to calculate the residual within the prior maps. This approach significantly enhances localization accuracy while maintaining processing speed. The overall design, as illustrated in Figure 1, is divided into four parts:Data processing for LiDAR and IMU: the initial pose estimation of the current frame is obtained through the preintegration of the IMU measurements. Subsequently, high-quality planar and ground points are extracted using a bidirectional projection plane slope difference filter. The IMU pre-integration results are propagated backwards to compensate for the motion distortion of LiDAR points in the current frame.Tightly coupled LiDAR-IMU odometry: the error equations for prediction and observation with ground constraints in the iterated extended Kalman filter (IEKF) are formulated using the points information and the pre-integration results from the IMU. As part of the estimation process, the pose is cyclically and iteratively updated.Relocation: the initial frame captured by the LiDAR is detected and corrected using prior historical keyframes for relocation. The initial frame is registered with the successfully detected historical keyframe, enabling the acquisition of accurate UGV position information within a prior map.Prior map maintenance: through the real-time maintenance of the three-layer voxels comprising the overall ROM voxels, nearest-neighbor RAM voxels, and ikd-tree voxels, the prior maps can be efficiently maintained, enabling a fast KNN search while ensuring lightweight operations.

## 3. Tightly Coupled LiDAR-IMU Odometry

### 3.1. Feature Extraction Based on Bidirectional Projection Plane Slope Difference Filter

The current most popular curvature feature extraction method involves calculating the relationship between each point and neighboring points on the local plane along the current LiDAR scanning line. However, this method suffers from relatively slow extraction speed and limited accuracy. Moreover, the curvature feature extraction method roughly classifies feature points into planar and edge points, but fails to account for the distinctive characteristics of ground points on the LiDAR scanning lines, leading to poor performance in extracting stable ground points and potential height value drift. Additionally, the curvature method exhibits a limited ability to filter out noisy and unstable points, resulting in potential lateral drift. Considering the scanning characteristics of 3D LiDAR, each scanning line is captured at a specific angle. When a line scans a plane, the points do not lie on a straight line in three-dimensional space. This observation makes feature extraction via projection planes more reliable. To address these limitations, this paper proposes a feature extraction method based on the bidirectional projection plane slope difference filter. This method only requires calculating the relationship between a point on the current line and its preceding and subsequent points. By mapping the calculation from three-dimensional space to two planes, the proposed method can extract high-quality ground, planar, and edge points simultaneously. Additionally, special processing is applied to lines in the proximity of the ground, enhancing both extraction accuracy and speed. The main steps of the proposed method are as follows.

(1) Near-end ground feature point extraction: the point cloud of the current frame is denoted as *P*. Leveraging the characteristics of 3D LiDAR, it is observed that LiDAR points from lower scanning lines tend to contain more stable ground features. This method establishes the lowest LiDAR scanning line $P_{min}$ as the reference line for near-end ground feature constraints. The line is divided into n segments based on the sector area, and the average distance from the LiDAR center to the LiDAR points in each segment is calculated. The segment $P_{ground}$ with the highest average distance is selected as the constraint for ground feature points. The average height $z_{ground}$ of these selected points represents an approximate estimation of the ground plane height in the near-end region. The lines ranging from $P_{min}$ to $P_{h}$, which are in proximity to the lowest scanning line, are identified as the near-end ground lines. On the other hand, the lines from $P_{h}$ to $P_{max}$ are classified as non-near-end ground lines. To mitigate the influence of noise and obstacles in the vicinity of the ground plane, points with heights, $z_{s}^{i}$, of the point *i* in the LiDAR scanning s-line falling outside the range [$z_{ground}$− u, $z_{ground}$ + u] within the near-end ground lines are filtered out, where u is the floating range threshold of the near-end ground height. The value of u is set according to the strictness of the near-end ground point filtering. This process facilitates the extraction of high-quality near-end ground points, thus enhancing the reliability of ground feature constraints for pose estimation.

(2) Far-end feature point extraction: this method involves the projection of points from each LiDAR scanning line, excluding those within the near-end ground lines of the current frame, onto the *l*-plane comprising the *x*-axis and *y*-axis of the LiDAR coordinate system. Simultaneously, the points are projected onto the *h*-plane, which consists of the *z*-axis and the angle θ within the plane formed by the *x*-axis and *y*-axis. Subsequently, the method applies a slope difference filter to each LiDAR scanning line on the two projection planes. The calculations performed in three-dimensional space are projected onto these two two-dimensional planes via the bidirectional projection plane slope filter. By evaluating the slope difference of the same LiDAR point on the two projection planes, the smoothness of the point on the local plane can be determined. This process allows for the efficient extraction of high-quality ground points, planar points, and edge points. The slope and slope difference of points in the *l*-plane and *h*-plane can be defined as follows: (1)$$K_{i}^{l}=\frac{\rho_{i+1}-\rho_{i}}{\rho_{i}\Delta \theta_{i}}$$
(2)$$d_{i}^{l}=K_{i}^{l}-K_{i-1}^{l}$$
(3)$$K_{i}^{h}=\frac{z_{i+1}-z_{i}}{\Delta \theta_{i}}$$
(4)$$d_{i}^{h}=\arccos\left(\frac{\Delta \theta_{i}\Delta \theta_{i-1}+\left(\Delta \theta_{i}K_{i}^{h}\right)\left(\Delta \theta_{i-1}K_{i-1}^{h}\right)}{\sqrt{{\left(\Delta \theta_{i}K_{i}^{h}\right)}^{2}+{\left(\Delta \theta_{i}\right)}^{2}}\sqrt{{\left(\Delta \theta_{i-1}K_{i-1}^{h}\right)}^{2}+{\left(\Delta \theta_{i-1}\right)}^{2}}}\right)$$
(5)$$\Delta \theta_{i}=\theta_{i+1}-\theta_{i}$$

In the formula, $K_{i}^{l}$ and $d_{i}^{l}$ are the slope and slope difference of point *i* on the *l*-plane. $K_{i}^{h}$ and $d_{i}^{h}$ are the slopes and slope difference of point *i* on the *h*-plane. $\rho_{i}$ is the distance between point *i* on the *l*-plane and the LiDAR center. $z_{i}$ is the height value of point *i*. $\theta_{i}$ is the angle value of point *i* on the *l*-plane. $\Delta \theta_{i}$ is the difference in angle between point i+1 and point *i* on the *l*-plane. As shown in Figure 2, the extraction effect of a point is shown. The near-end ground points, high-quality planar points and high-quality ground points are red, and the corner points are green.

The slope difference filter formulas enable the evaluation of whether the current point, along with its preceding and succeeding points on the same line, can be fitted into a straight line in the *l*-plane or *h*-plane. A close approximation to a straight line in the *l*-plane indicates that the point lies on a non-ground plane within three-dimensional space. The projection of ground points onto the *l*-plane exhibits a circular arc shape. Conversely, an approach to a straight line in the *h*-plane suggests that there is no significant change in the *z*-axis height between the points. When the slope difference on the *l*-plane approaches zero and the slope difference on the *h*-plane is less than the threshold value for a jump on the *h*-plane, it indicates that the point is a stable planar point. The constraint on the *l*-plane serves to filter out noise and unstable points. When the slope difference on the *l*-plane approaches zero and the slope difference on the *h*-plane is greater than the jump threshold, it suggests that the angle between the laser beam of the current point and the plane is small, indicating that the point is located at the far end of the plane. These points are considered unstable planar far-end points that need to be filtered out. When the slope difference on the *l*-plane cannot approach zero and the slope difference on the *h*-plane does not approach zero but is less than the jump threshold, it indicates that the point is a corner point. The constraint on the *h*-plane serves to filter out the far jump points. When the slope difference on the *l*-plane cannot approach zero and the slope difference on the *h*-plane is greater than the jump threshold, it suggests that the point transitions from one plane to another far plane. Lastly, when the slope difference on the *l*-plane cannot approach zero and the slope difference on the *h*-plane strictly approaches zero, it indicates that the point is a stable ground point.

Algorithm 1 outlines the feature extraction method in this paper. $d_{i}^{l}$ is the slope difference on the *l*-plane of the point *i* in the current frame. $d_{i}^{h}$ is the slope difference on the h-plane of the point *i* in the current frame. $d_{lm}$ is the threshold at which the slope difference of the point on the *l*-plane approaches zero. $d_{hn}$ is the threshold at which the slope difference of the point on the *h*-plane strictly approaches zero. $d_{hm}$ is the threshold at which the slope difference of a point on the *h*-plane does not jump. ${\bar{P}}_{p}$ is a high-quality planar point, ${\bar{P}}_{d}$ is a high-quality ground point, and ${\bar{P}}_{d}$ is a high-quality edge point.    
**Algorithm** **1:** Feature extraction
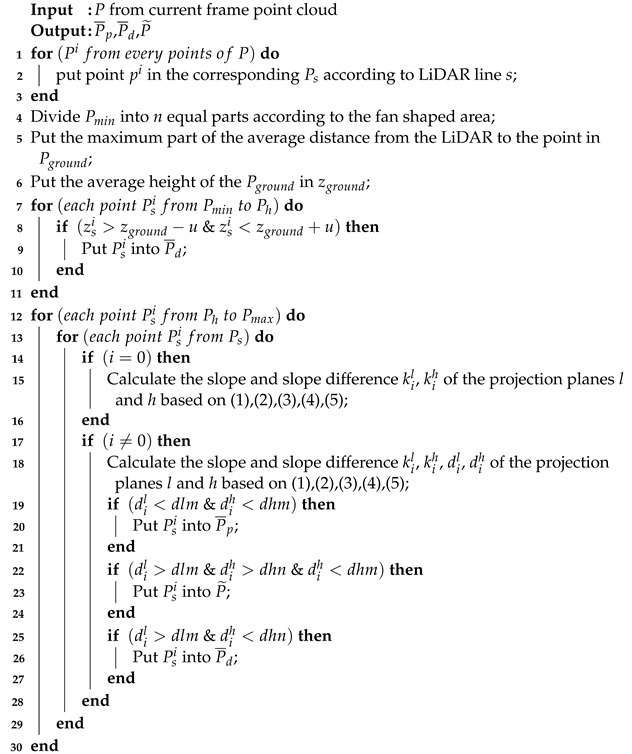


### 3.2. IMU Preintegration and Compensation of Point Cloud Distortion

The IMU preintegration and compensation of point cloud distortion are performed using discretized forward and backward propagation. The forward propagation is conducted over the time interval $\Delta t_{f}$ between consecutive IMU frames. It begins with the IMU measurement taken after the LiDAR point timestamp at the start of the current frame and ends with the IMU measurement taken before the LiDAR point timestamp at the end of the current frame. On the other hand, backward propagation compensates for point cloud distortion in the current frame based on the time difference $\Delta t_{l}$ between adjacent LiDAR scanning points.

(1) Forward propagation: in the continuous model, the position, velocity and attitude of the IMU can be defined as follows: (6)$$v_{b}^{G}=v_{a}^{G}+\int_{t\epsilon [a,b]}R_{t}^{G}\left(a_{t}-b_{at}-n_{at}\right)+g^{G}dt$$
(7)$$p_{b}^{G}=p_{a}^{G}+\int_{t\epsilon [a,b]}v_{t}+R_{t}^{G}\left(a_{t}-b_{at}-n_{at}\right)t+g^{G}tdt$$
(8)$$R_{b}^{G}=R_{a}^{G}exp\left(\int_{t\epsilon [a,b]}\omega_{t}-b_{\omega t}-n_{\omega t}dt\right)$$

In the equation, $v_{a}^{G}$ and $v_{b}^{G}$ are the velocities of IMU at time *a* and *b* in the world coordinate system. $p_{a}^{G}$ and $p_{b}^{G}$ are the positions of IMU at time *a* and *b* in the world coordinate system. $R_{a}^{G}$, $R_{b}^{G}$ and $R_{t}^{G}$ are the attitude of the IMU at times *a*, *b*, and *t* in the world coordinate system. $a_{t}$ is the acceleration measured at time *t* in the IMU coordinate system. $v_{t}$ is the speed at time *t* in the IMU coordinate system. $\omega_{t}$ is the angular velocity measured at time *t* in the IMU coordinate system. $g^{G}$ is the gravitational component in the world coordinate system. $b_{at}$ and $n_{at}$ are acceleration bias and noise at time *t*. $b_{\omega t}$ and $n_{\omega t}$ are the angular velocity bias and noise at time *t*. After removing the unknown noise and offsetting, discretizing and simplifying it, the IMU forward propagation model is defined as follows: (9)$$v_{k+1}=v_{k}+\left(R_{k}^{G}a_{k}+g^{G}\right)\Delta t_{f}$$
(10)$$p_{k+1}=p_{k}+v_{k}\Delta t_{f}$$
(11)$$R_{k+1}=R_{k}exp\left(\omega_{k}\Delta t_{f}\right)$$
(12)$$v_{0}={\bar{v}}_{i}$$
(13)$$p_{0}={\bar{p}}_{i}$$
(14)$$R_{0}={\bar{R}}_{i}$$

In the formula, $v_{k}$, $v_{k+1}$ and $v_{0}$ are the speeds of the current frame IMU at times *k*, k+1, and 0 in the world coordinate system. $p_{k}$, $p_{k+1}$ and $p_{0}$ are the positions of the current frame IMU at times *k*, k+1, and 0 in the world coordinate system. $R_{k}$, $R_{k+1}$ and $R_{0}$ are the attitude of the IMU at times *k*, k+1 and 0 in the world coordinate system. $a_{k}$ is the accelerometer measurement at time *k* in the IMU coordinate system of the current frame. $\omega_{k}$ is the angular velocity measurement at time *k* in the IMU coordinate system of the current frame. ${\bar{v}}_{i}$, ${\bar{p}}_{i}$ and ${\bar{R}}_{i}$ are the true value of velocity, position and rotation matrix of the last frame after the Kalman filter. When the time k+1 is the time of the last LiDAR point in the current frame, $v_{k+1}$, $p_{k+1}$ and $R_{k+1}$ are the ${\hat{v}}_{i+1}$, ${\hat{p}}_{i+1}$ and ${\hat{R}}_{i+1}$ entering the Kalman filter at the end of the current frame i+1.

(2) Backward propagation: similar to the discretized forward propagation, the backward propagation propagates the IMU measurements in the opposite time direction -$\Delta t_{l}$ to compensate for the distortion of points in the current frame. This process occurs at the frequency of LiDAR scanning points. Since the IMU measurement frequency is lower than the LiDAR scanning frequency, there are multiple LiDAR points between two consecutive IMU measurements. Consequently, the most recent IMU measurement prior to the timestamp of the current LiDAR point is utilized as the input value for backward propagation. The IMU backward propagation model is defined as follows:(15)$$v_{b}=v_{b+1}-\Delta t_{l}\left(R_{b+1}^{G}a_{k}+g^{G}\right)$$
(16)$$p_{b}=p_{b+1}-\Delta t_{l}v_{b+1}$$
(17)$$R_{b}=log\left({\left(\omega_{k}\Delta t_{l}\right)}^{T}R_{b+1}\right)$$

$v_{b}$, $p_{b}$ and $R_{b}$ represent the velocity, position, and attitude of the b-th point in the current LiDAR scan frame in the world coordinate system. $a_{k}$ and $\omega_{k}$ are the acceleration and angular velocity obtained from the closest IMU measurement taken prior to the timestamp of the b+1-th point in the current LiDAR scan frame. When time b+1 is the time of the last LiDAR point in the current frame, $v_{b+1}$, $p_{b+1}$ and $R_{b+1}$ are the initial inputs ${\hat{v}}_{i+1}$, ${\hat{p}}_{i+1}$ and ${\hat{R}}_{i+1}$ for the backward propagation in the current frame i+1.

### 3.3. Back-End Optimization Based on Iterated Extended Kalman Filter

#### 3.3.1. Error State Estimation

To facilitate error calculation and state update, the state of UGV is defined as $[R^{G^{T}},p^{G^{T}},v^{G^{T}},b_{\omega }^{T},b_{a}^{T},g^{G^{T}}{]}^{T}$, and the error ε between the estimated state and the actual state is defined as ${\left[\varepsilon^{R^{T}},\varepsilon^{p^{T}},\varepsilon^{v^{T}},\varepsilon^{b\omega^{T}},\varepsilon^{ba^{T}},\varepsilon^{g^{T}}\right]}^{T}$. The integrated state can be expressed as $[{\left(R^{G}exp\left[\varepsilon^{R}\right]\right)}^{T},(p^{G}+\varepsilon^{p}{)}^{T},(v^{G}+\varepsilon^{v}{{)}^{T},{\left(b_{\omega }+\varepsilon^{b_{\omega }}\right)}^{T},{\left(b_{a}+,\varepsilon^{b_{a}}\right)}^{T},{\left(g^{G}+\varepsilon^{g}\right)}^{T}]}^{T}$. In the formula, $\varepsilon^{R}$ represents the error in the state rotation matrix. $\varepsilon^{p}$ is the error in the state position. $\varepsilon^{v}$ is the error in the state velocity. $\varepsilon^{b_{\omega }}$ is the error in the angular velocity bias. $\varepsilon^{b_{a}}$ is the error in the accelerometer bias error.

(1) Predicted error state: the predicted error is defined as the discrepancy between the real state and the propagated state. By utilizing the propagated formula of the real state and the propagated formula of the propagated state, the propagated formula of the predicted error state and the covariance propagation formula are defined as follows: (18)$${\hat{\varepsilon }}_{i+1}=F_{{\hat{\varepsilon }}_{i}}\Delta t_{f}{\hat{\varepsilon }}_{i}+{\hat{\varepsilon }}_{i}$$
(19)$${\hat{P}}_{i+1}=F_{{\hat{\varepsilon }}_{i}}{\hat{P}}_{i}F_{{\hat{\varepsilon }}_{i}}^{T}+F_{\omega_{i}}QF_{\omega_{i}}^{T}$$

In the formula, ${\hat{\varepsilon }}_{i}$ is the predicted error state at time *i*. $\omega_{i}$ is the Gaussian white noise at time *i*. $F_{{\hat{\varepsilon }}_{i}}$ is the error state propagated matrix at time *i*. $F_{\omega_{i}}$ is the Jacobian matrix of the Gaussian white noise at time *i*. ${\hat{P}}_{i}$ and *Q* are the covariance of ${\hat{\varepsilon }}_{i}$ and $\omega_{i}$ at time *i*. $F_{{\hat{\varepsilon }}_{i}}$ and $F_{\omega_{i}}$ are more concretely derived in [18].

After obtaining the estimation of the propagated error state, the predicted error state of the iterated update is defined as follows:(20)$${\tilde{\varepsilon }}_{k}={\hat{\varepsilon }}_{k}^{a}+h_{a}{\tilde{\varepsilon }}_{k}^{a}$$
(21)$$P_{k}^{a+1}={\left(h_{a}\right)}^{-1}P_{k}^{a}{\left(h_{a}\right)}^{-T}$$

In the equation, ${\tilde{\varepsilon }}_{k}$ is the error between the true state of the frame *k* and the propagated state of frame *k*. ${\hat{\varepsilon }}_{k}^{a}$ is the error between the *a*-th update state of the frame *k* and the propagated state of the frame *k*. $P_{k}^{a}$ is the covariance of the *a*-th state update of the frame *k*. $h_{a}$ is the Jacobian matrix of ${\tilde{\varepsilon }}_{k}$ relative to ${\tilde{\varepsilon }}_{k}^{a}$.

(2) Observed error state: the LiDAR point cloud of the current frame is projected onto the global coordinate system using IMU preintegration. The residual distance between the projected points of the current frame and the nearest local plane of the prior map is defined as the observed state error. The observed error state is defined as follows: (22)$$0=z_{j}=S_{J}\left[\left(R_{k}M_{j}+P_{k}\right)-M_{j}^{G}\right]\approx z_{j}^{a}+H_{j}^{k}{\bar{\varepsilon }}_{k}^{a}$$
(23)$$z_{j}^{a}=S_{J}\left[\left(R_{k}^{a}M_{j}+P_{k}^{a}\right)-M_{j}^{G}\right]$$

In the formula, $z_{j}^{a}$ is the residual distance between point *j* of the *a*-th update state of frame *k* in state ${\hat{x}}_{k}^{a}$ and the nearest local plane in the prior map. $z_{j}$ is the residual distance between point *j* of the current frame *k* in state $x_{k}$ and the nearest local plane in the prior map. $S_{J}$ is the normal vector of the local plane fitted by the nearest neighbor points of point *j* in the current frame *k* within the prior map. $M_{j}$ is the coordinate of point *j* in the coordinate system of the current frame *k*. $R_{k}^{a}$ and $P_{k}^{a}$ are the rotation matrix and translation vector of state ${\hat{x}}_{k}^{a}$. $R_{k}$ and $P_{k}$ are the rotation matrix and translation vector of state $x_{k}$. $M_{j}^{G}$ is the nearest-neighbor point of point *j* in the prior map of the current frame *k*. $x_{k}$ and ${\hat{x}}_{k}^{a}$ represent the true state of frame *k* and the *a*-th update state of frame *k*. ${\bar{\varepsilon }}_{k}^{a}$ is the error between the true state and the *a*-th update state of frame *k*. $H_{k}^{a}$ is the Jacobian matrix of $z_{j}$ relative to ${\bar{\varepsilon }}_{k}^{a}$. The nearest neighbor points are obtained through KNN search in the ikd-tree.

#### 3.3.2. Iterated State Update with Ground Constraints Added

Combining the predicted error state in (24) and the observed error state in (27) yields the maximum posterior estimate (MAP). To address the limitations posed by a limited number of points on the ground or plane, a new MAP formula is proposed, which incorporates ground constraints and utilizes a feature extraction method to accurately divide high-quality ground and planar points. The proposed method assigns adaptive weights to the observed error equations of plane and ground points. Specifically, a higher weight is assigned to the observed error equation of ground points when there are more planar points, and a higher weight is assigned to the observed error equation of planar points when there are more ground points. By employing this new formula, the insufficiency of lateral or longitudinal constraints caused by a scarcity of points on the ground or plane can be reduced. The new formula is defined as follows: (24)$$\underset{{\bar{\varepsilon }}_{k}^{a}}{\min}\left({∥\begin{array}{c} {\hat{\varepsilon }}_{k}^{a}+h_{a}{\bar{\varepsilon }}_{k}^{a} \end{array}∥}_{P{}_{k}^{a^{-}1}}^{2}+\frac{m+n}{2n}\sum_{j=1}^{n}{∥\begin{array}{c} z_{j}^{a}+H_{j}^{a}{\bar{\varepsilon }}_{k}^{a} \end{array}∥}_{U{}_{k}^{a^{-}1}}^{2}+\frac{m+n}{2m}\sum_{j=n+1}^{n+m}{∥\begin{array}{c} z_{j}^{a}+H_{j}^{a}{\bar{\varepsilon }}_{k}^{a} \end{array}∥}_{U{}_{k}^{a^{-}1}}^{2}\right)$$

In the equation, ${∥\begin{array}{c} x \end{array}∥}_{p}^{2}=x^{T}Px$. $U_{k}^{a}$ is the covariance of the observed noise for the *a*-th state update in the frame *k*. *n* is the number of planar points in current frame *k*. m is the number of ground points in current frame *k*. According to references [33,34], the Kalman gain $K_{k}^{a}$ of the *k*-th frame and the update error state ${\bar{\bar{\varepsilon }}}_{a}^{a+1}$ of the *a*-th iteration in the frame *k* after dimensions are reduced are defined as follows: (25)$$K_{k}^{a}={\left(HU_{k}^{a}H+P^{-1}\right)}^{-1}H^{T}U_{k}^{a^{-1}}$$
(26)$${\bar{\bar{\varepsilon }}}_{a}^{a+1}=-K_{k}^{a}z_{k}^{a}-\left(I-K_{k}^{a}H\right)h_{a}^{-1}{\hat{\varepsilon }}_{k}^{a}$$

If *n* is greater than *m*, $H={\left[H_{1}^{a},\cdots ,H_{n}^{a^{T}},H_{n+1}^{a^{T}},\cdots ,H_{n+m}^{a^{T}},\cdots ,H_{2n}^{a^{T}}\right]}^{T}$, $R=diag(R_{1},$$\cdots ,R_{n},R_{n+1},\cdots ,R_{n+m},\cdots ,R_{2n})$, $z^{a}={\left[z_{1}^{a^{T}},\cdots ,z_{n}^{a},z_{n+1}^{a^{T}},\cdots ,z_{n+m}^{a^{T}},\cdots ,z_{2n}^{a^{T}}\right]}^{T}$. $[H_{n+m}^{a^{T}},$$\cdots ,H_{2n}^{a^{T}}]$, $\left[R_{n+m},\cdots ,R_{2n}\right]$, $\left[z_{n+m}^{a^{T}},\cdots ,z_{2n}^{a^{T}}\right]$ is the pseudo occupancies of ground constraints reusing high-quality ground points. If *m* is greater than *n*, $H=[H_{1}^{a},\cdots ,H_{n}^{a^{T}},$$H_{n+1}^{a^{T}},\cdots ,$$H_{m}^{a^{T}},\cdots ,H_{2m}^{a^{T}}{]}^{T}$, $R=diag(R_{1},\cdots ,R_{n},R_{n+1},\cdots ,R_{m},\cdots ,R_{2m})$, $z^{a}=[z_{1}^{a^{T}},\cdots ,z_{n}^{a},z_{n+1}^{a^{T}},$$\cdots ,z_{m}^{a^{T}},\cdots ,z_{2m}^{a^{T}}{]}^{T}$. $\left[H_{n+1}^{a^{T}},\cdots ,H_{m}^{a^{T}}\right]$, $\left[R_{n+1},\cdots ,R_{m}\right]$, $\left[z_{n+1}^{a^{T}},\cdots ,z_{m}^{a^{T}}\right]$ is the pseudo occupancies of planar constraints reusing high-quality planar points. By combining the states of ${\hat{x}}_{k}^{a}$ and ${\bar{\bar{\varepsilon }}}_{a}^{a+1}$, the updated state ${\hat{x}}_{k}^{a+1}$ of time a+1 can be obtained. When the error between ${\hat{x}}_{k}^{a+1}$ and ${\hat{x}}_{k}^{a}$ is small enough, the iteration will be terminated. The final state ${\bar{x}}_{k}$ and covariance matrix ${\bar{P}}_{k}$ of the frame *k* are defined as follows: (27)$${\bar{x}}_{k}={\hat{x}}_{k}^{a+1}$$
(28)$${\bar{P}}_{k}=\left(I-KH\right)P$$

## 4. Relocation

To obtain the prior pose of UGV in the prior map, a relocation method is devised by combining detection and correction with the iterative closest point (ICP) method. This method is integrated into the front-end odometry to initialize the UGV’s pose. The proposed method involves extracting keyframes from significant positions within the prior map, forming prior keyframes. Following the method described in reference [35], we generate descriptor matrices for the current frame and historical keyframes. By employing a two-stage matching approach, candidate relocalization keyframes for the current frame are identified through the column vectors formed using ring key encoding information. Rotation and translation offsets are calculated through the row summation of these column vectors. Subsequently, the descriptor of the current frame is matched with the descriptor matrix of the candidate relocalization keyframes assigned with the computed offsets, resulting in the identification of the relocalization keyframe. A rough correction is then applied to the UGV’s current attitude using the calculated angle offset. Finally, an accurate UGV pose within the prior map is obtained by performing ICP point cloud registration on the corrected initial frame and the relocation keyframe. To address the limitations associated with traditional point-to-point, point-to-plane, and linear least-squares optimization methods for point-to-plane ICP [36,37,38], which involve multiple iterations, time consumption, and are prone to overfitting, the generalized ICP (G-ICP) [39] is employed for point cloud registration. This method reduces the number of iterations while improving matching accuracy.

## 5. Prior Map Maintenance

The observed equation requires identifying the nearest neighboring points in the prior map to perform residual fitting. To achieve a fast and stable KNN search on UGVs with limited RAM and CPU capabilities, the prior map is maintained by combining the proposed three-layer voxels and the ikd-tree. This method ensures efficient and reliable execution, as shown in Figure 3.

First, the prior map is voxelized based on the position of the world coordinate system and stored in ROM accordingly. The voxel files are named [sign(x),100|x|+|y|,sign(y)] according to the voxel positions in a prior map coordinate system, forming the first-layer ROM voxels. By retrieving file names containing position information, the voxels near the current position, obtained from the relocation, are extracted from the system ROM into the corresponding voxel container in RAM, forming the second-layer RAM voxels. The point clouds within the nearest nine RAM voxel containers to the current position are loaded into the ikd-tree, forming the third-layer ikd-tree voxels. This method reduces the number of point clouds in RAM and the size of the ikd-tree, effectively minimizing RAM consumption, and enabling UGV to perform real-time localization using a large prior map. Once the initialization of prior map maintenance is completed, the map is continuously updated in real time. Real-time updates are performed on the second and third layers of voxels using the 6DOF state estimation output from the odometry. When the LiDAR center moves to another voxel range, this method removes the point clouds outside the ikd-tree voxel range of the LiDAR center in the ikd-tree, and inserts the point clouds of the newly added ikd-tree voxels from the RAM voxels into the ikd-tree. Simultaneously, the RAM containers outside the RAM voxel range of the LiDAR center at the corresponding position are emptied. The newly added RAM voxels are extracted from the corresponding ROM voxel into the corresponding RAM voxel container at the respective position. To ensure independent operation and avoid interference between the map maintenance modules of the second and third layers, as well as to prevent any impact on the real-time KNN search of the odometry, this method runs the RAM voxel maintenance, ikd-tree voxel maintenance, and odometry in three separate threads. This setup improves the system’s real-time performance. Additionally, by setting RAM voxels and ikd-tree voxels separately, the real-time performance of inserting voxel point clouds into the ikd-tree is enhanced, thereby accelerating the operating cycle of the odometry. The process of prior map maintenance is summarized in Algorithm 2.
**Algorithm** **2:** Map maintenance
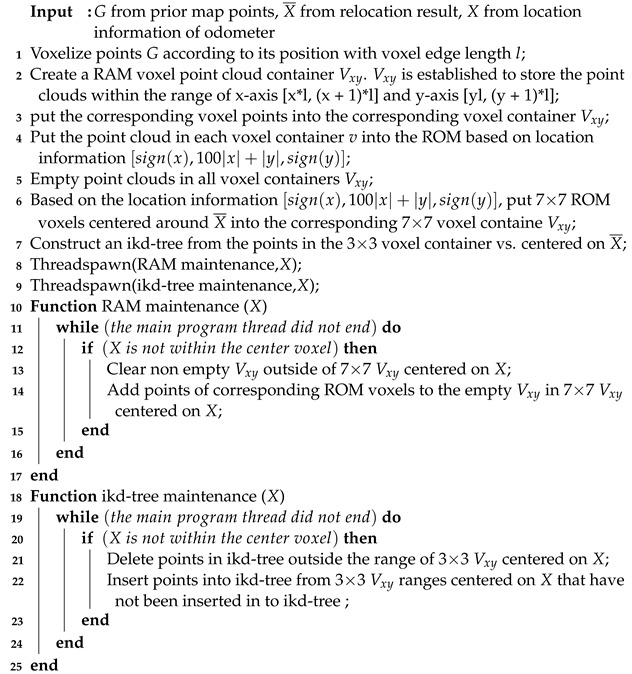


## 6. Experiment and Analysis

In this experimental section, we describe our ablation experiments, which we conducted to analyze the accuracy and efficiency of each proposed component separately and compare them with advanced SLAM methods. Adhering to the principles of fairness and impartiality, all comparative experiments were conducted in the same environment, and the experimental data were both accurate and reliable. The experimental hardware platform consisted of a laptop, a wheeled UGV, a Velodyne VLP-16 LiDAR, and an Xsens MTI-100 IMU, as depicted in Figure 4. The Velodyne VLP-16 LiDAR mounted on the experimental platform had a measurement range of 100 m, a vertical viewing angle ranging from +15° to −15°, a horizontal viewing angle of 360°, and vertical and horizontal resolutions of 2° and 0.1° to 0.4°, respectively. It offered a measurement accuracy of ±3 cm and operated at a frequency of 20 Hz. The Xsens MTI-100 IMU incorporated a built-in gyroscope, accelerometer, and magnetometer, with a sampling frequency of up to 10 KHz. The laptop utilized in the experiments featured an Intel (R) Core (TM) i7-12700H processor, operating at a frequency of 2.3 GHz, with 16 GB of RAM. The software system ran on Ubuntu 18.04 and employed ROS Melodic [40] as the middleware. The algorithm was implemented in C++.

To evaluate the performance of each module, experiments were conducted using self-collected indoor and outdoor closed-loop datasets from an industrial park, as well as publicly available KITTI datasets. The self-collected datasets were acquired within a standardized industrial park environment. The outdoor dataset encompasses notable structures within the industrial park, including industrial factories, vegetation, parking lots, office buildings, and roads. The trajectory length of this dataset is approximately 308 m. The indoor dataset consists of distinctive structures such as columns, beams, walls, and industrial equipment, with a trajectory length of approximately 202 m. It is important to note that both the indoor and outdoor datasets were meticulously designed to commence and conclude at the same position during data collection. To bolster the credibility of the experiments, the odometry performance was also assessed using the publicly available KITTI dataset. This dataset was generated by equipping a car with various sensors and driving along well-structured roads. The car’s setup included a Velodyne HDL-64E LiDAR, Oxts R3003 IMU, two grayscale cameras, two color cameras, four optical lenses, and a GPS navigation system.

To assess the accuracy and efficiency of the proposed method, trajectory maps for both the proposed method and other existing methods are generated. Comparative experiments are conducted by calculating the time consumption of each module. The corresponding errors, including trajectory error, absolute pose error, and closed-loop error, are computed. Additionally, to evaluate the global consistency of each method’s trajectory, the root mean square error (RMSE) is calculated for each trajectory. Moreover, time consumption curves for each component are plotted to facilitate a more intuitive analysis of the method’s efficiency.

## 7. Analysis of Mapping Results without Prior Maps

To evaluate the efficiency and accuracy of the proposed feature extraction method and odometry method, a horizontal comparison was performed. The proposed method was compared with FAST-LIO2 and FAST-LIO2 combined with the curvature feature extraction method. To ensure reliability and credibility, the experiments were conducted using the KITTI dataset. The evaluation tool EVO [41] was employed to visualize the experimental trajectories and ground truth trajectory, and to compute the absolute pose error for accuracy analysis. Furthermore, the number of feature points per frame and the time consumption curves of each component were plotted to analyze the efficiency.

### 7.1. Analysis of Mapping Accuracy without Prior Maps

Experiments were conducted on the KITTI dataset using three different methods: FAST-LIO2, FAST-LIO2 combined with curvature feature extraction, and the proposed odometry method combined with the bidirectional projection plane slope difference filter feature extraction. The results and trajectories obtained from these experiments are presented in Figure 5 and Figure 6, respectively.

Figure 5 depicts the top view and local details of the running results obtained from the three methods. The top view (a, b, c) and the local detail view (d, e, f) of Figure 5 provide a visual comparison of the mapping effects of the three methods. Both FSAT-LIO2 and FAST-LIO2 combined with the curvature feature extraction method exhibit significant drift, particularly at turns and certain positions where feature degradation occurs in space. Notably, the FAST-LIO2 combined with the curvature feature extraction method demonstrates severe vertical height drift. In contrast, the proposed odometry method, combined with the proposed feature extraction method, effectively suppresses drift during turns, positions of feature degradation, and vertical heights. Furthermore, it exhibits minimal translocation at revisited positions. The aforementioned effect can be attributed to the curvature feature extraction method’s inability to accurately extract stable ground points, leading to insufficient constraints on vertical height and significant vertical height drift. Moreover, the curvature extraction method fails to filter out noise and unstable points during plane feature extraction, resulting in insufficient lateral constraints and some degree of drift in the lateral direction. The method that does not perform feature extraction utilizes a large number of original points for residual calculation. While its vertical height constraints are superior to the curvature method, it fails to filter out unstable points and noise within the point clouds. When the proportion of noise and unstable points cannot be ignored, it affects the effectiveness of residual fitting, resulting in significant drift. In contrast, the proposed method strengthens ground constraints by filtering ground points near the ground and extracting high-quality ground and planar points from the far end using the bidirectional projection plane slope difference filter. It incorporates ground constraints into the back-end optimization formulation, imposing strict constraints on both vertical height and horizontal plane, which yields outstanding mapping results. Figure 6 presents the 3D trajectory, horizontal planar trajectory, vertical planar trajectory, and three-axis trajectory generated by the EVO for the three methods. Comparing the accuracy of odometry among the three methods, the proposed odometry method combined with the proposed feature extraction method demonstrates the trajectory that most closely aligns with the true trajectory. Additionally, it exhibits the best closed-loop effect at the end. In contrast, FSAT-LIO2 and FAST-LIO2 combined with the curvature feature extraction method result in significant translocation at the end. Furthermore, FAST-LIO2 combined with the curvature feature extraction method exhibits severe vertical height drift. The results depicted in Figure 6 validate the accuracy and efficiency of the proposed method, confirming the advantages and feasibility of both the proposed feature extraction method and odometry method.

To assess the global consistency of the three methods and quantitatively evaluate their errors, we calculate the absolute pose error (APE) between their respective trajectories and the ground truth trajectory. The calculated APE values are presented in Table 1.

When testing on the KITTI dataset, the root mean square error (RMSE) of the absolute pose error obtained using the proposed odometry method combined with the proposed feature extraction method exhibits a 48.24% increase compared to FAST-LIO2 combined with the curvature feature extraction method. Furthermore, the RMSE of the proposed method is 30.19% higher than that of FAST-LIO2. These results demonstrate that the proposed odometry method combined with the proposed feature extraction method achieves superior performance.

### 7.2. Analysis of Mapping Efficiency without Prior Maps

This paper presents a comprehensive analysis of three methods by plotting the number of feature points, the time consumption for feature extraction, and the time consumption for odometry in each frame. Mean values are calculated to provide a summary of the results. Through a visual comparison of feature point counts and time consumption across different method components, the impact of the proposed feature extraction method and odometry method on time consumption and computational complexity is analyzed. The curves depicting the number of feature points and time consumption for each component are illustrated in Figure 7. Furthermore, Table 2 presents the average number of feature points across all frames and the average time consumption for each component.

The feature point count, feature extraction time, and odometry time of the three methods per frame are compared in Figure 7a–c. The average time consumption and average number of feature points are presented in Table 2. To analyze the improvement in odometry efficiency offered by the proposed method, a comparison is made between the time consumption and calculated amount of the three methods. The proposed feature extraction method reduces the average number of feature points extracted per frame by 39.70% compared to the curvature feature extraction method. Furthermore, it achieves an 83.01% reduction in feature points compared to the method without feature extraction. Consequently, the proposed method achieves higher odometry accuracy with fewer feature points. In terms of time consumption, the proposed feature extraction method reduces the average feature extraction time by 36.07% compared to the curvature feature extraction method. When the proposed feature extraction method is combined with the proposed odometry method, it reduces the average odometry time by 42.62% compared to FAST-LIO2 combined with the curvature feature extraction method. Additionally, the proposed method reduces the average odometry time by 74.81% compared to FAST-LIO2. This reduction in time consumption is due to the proposed method’s projection of 3D space onto a 2D plane, which requires less calculation for feature extraction compared to 3D space feature extraction methods. Overall, the proposed method reduces the overall time consumption per frame by 40.06% compared to FAST-LIO2 combined with the curvature method and achieves a 56.72% reduction compared to FAST-LIO2 alone. These findings demonstrate that the proposed method significantly improves the efficiency of the overall system, and therefore the real-time requirements for odometry processing are met.

## 8. Analysis of Localization Results Using Prior Maps

To assess the influence of prior maps on the UGV’s localization accuracy and the impact of the proposed prior map maintenance method on KNN search speed and RAM consumption, closed-loop experiments are conducted on self-collected datasets acquired from indoor and outdoor environments. In the case of utilizing prior maps, horizontal comparisons are performed using different methods. The located accuracy of each method is evaluated by analyzing the closed-loop errors along the three axes and the overall performance. The HTOP [42] is employed to calculate the average RAM consumption of the proposed map maintenance method and the alternative method, as well as to determine the average KNN search speed for both methods. Furthermore, this paper presents the time consumption means of individual components of odometry to analyze the efficiency and practicality of the proposed method.

### 8.1. Analysis of Localization Accuracy Using Prior Maps

Closed-loop experiments were conducted on self-collected indoor and outdoor datasets using FAST-LIO2 combined with the proposed prior map maintenance method, and proposed odometry method combined with the proposed prior map maintenance method and proposed feature extraction method. When utilizing the prior map for localization, the observed angle deviation of the closed-loop position in the outdoor results, as well as the angle and height deviation of the closed-loop position in the indoor results, are attributed to discrepancies between the prior map coordinate system and the odometry coordinate system. However, the computed closed-loop error values are accurate. Figure 8 illustrates the indoor and outdoor results, along with the results of the local closed-loop position. The closed-loop errors for all three axes and the overall errors are presented in Table 3.

The comparative analysis of the intuitive mapping effect and closed-loop error between the two methods is presented in the outdoor and indoor results, as well as in Figure 8 and Table 3. In the outdoor test scenario, when FAST-LIO2 is combined with the proposed prior map maintenance method, it is evident that the initial and final positions overlap, showcasing an outstanding closed-loop effect. The total closed-loop error measures 0.097 m, corresponding to a proportion of total mileage of 0.0315%. This achievement demonstrates a centimeter-level localization accuracy, validating the effectiveness of utilizing prior maps for localization. Moreover, when the proposed odometry method is combined with the proposed prior map maintenance method and the proposed feature extraction method, the initial and final positions also exhibit overlapping characteristics. The utilization of the proposed feature extraction method facilitates the filtration of unstable point clouds such as grass and cars, resulting in the extraction of high-quality ground points. Furthermore, it is observed that the extracted planar points exhibit clearer lines and boundaries compared to the other method, indicating a high quality of extracted planar points. The total closed-loop error is reduced to 0.044 m, with a proportion of total mileage of 0.0143%. This achievement signifies improved localization accuracy, with a smaller number of feature points. Remarkably, the localization accuracy along the Z-axis reaches the millimeter level. In the indoor test scenario, the overall closed-loop error for the combination of FAST-LIO2 and the proposed prior map maintenance method is measured at 0.029 m, with a proportion of total mileage at 0.0144%. The total closed-loop error for the proposed odometry method, combined with the proposed prior map maintenance method and the proposed feature extraction method, is reduced to 0.021 m, corresponding to a proportion of total mileage of 0.0104%. In both outdoor and indoor tests, the proposed odometry method, combined with the proposed prior map maintenance method and the proposed feature extraction method, achieves an improvement of 54.639% and 38.095% in closed-loop accuracy compared to the combination of FAST-LIO2 and the proposed prior map maintenance method. These findings highlight that localization using prior maps meets the required accuracy standards for UGVs in industrial parks. Additionally, the proposed method significantly enhances the localization accuracy of UGVs within industrial park environments.

### 8.2. Analysis of Localization Efficiency Using Prior Maps

To evaluate the localization efficiency of the proposed UGV located method in industrial parks, this paper computes the average time consumption for each segment of both indoor and outdoor datasets. The results for the average time consumption of each segment are presented in Table 4.

The average processing time for the proposed combination of FAST-LIO2 with the prior map maintenance method is 11.598 ms, and for the proposed odometry method with the prior map maintenance method and feature extraction method is 7.849 ms. The total processing speed of the proposed odometry method, when combined with the prior map maintenance method and feature extraction method, exhibits a 32.33% improvement compared to the combination of FAST-LIO2 with the prior map maintenance method. It can be observed that the proposed odometry method, in conjunction with the prior map maintenance method and feature extraction method, satisfies the requirements for UGV odometry processing speed in industrial parks, while ensuring sufficient localization accuracy. Therefore, it is more suitable for real-time UGV operations in industrial park environments.

### 8.3. Analysis of Effectiveness of Map Maintenance

To assess the practicality and validity of the map maintenance module, we employ the proposed map maintenance method alongside the approach of incorporating all point clouds from prior maps into the ikd-tree. This enables us to evaluate the associated RAM usage and KNN search speed. The computed results are presented in Table 5.

The prior map area utilized is 1320 m × 1320 m, consisting of approximately 12.45 million point clouds. The same number of point clouds, 12.45 million, are added to the ikd-tree. On average, the RAM usage amounts to 15 GB, occupying 98% of the total RAM. It is evident that the method of including all prior map point clouds in the ikd-tree exceeds the host’s RAM capacity, resulting in lag and data loss. Consequently, the system becomes unable to run when handling larger prior maps of industrial parks. In contrast, the proposed map maintenance method significantly reduces RAM consumption on the host. Specifically, the number of point clouds in the ikd-tree is reduced to 2.335 million, resulting in an average RAM usage of 5.2 GB, occupying 34% of the total RAM. This observation demonstrates the effectiveness of the proposed map maintenance method in reducing RAM consumption, thereby enabling the loading and maintenance of prior maps for large industrial parks. Since the search speed of the ikd-tree relies on the number of layers within the tree structure, and the relationship between the number of layers and the number of point clouds follows a power-of-two pattern, the KNN search speed of the proposed map maintenance method is only 10% higher than that of the method that includes all point clouds in the ikd-tree.

## 9. Conclusions

This paper presents a method for real-time UGV localization in industrial parks utilizing prior maps. This paper presents a novel feature point extraction method, improves the back-end optimization function of FAST-LIO2, and introduces a new prior map maintenance method. To enhance the quality and efficiency of feature extraction, a novel feature extraction method based on the bidirectional projection plane slope difference filter is proposed. This method improves extraction speed while maintaining extraction accuracy. It enables the separate extraction of ground and planar points, as well as edge points, using a unified method. The back-end optimization incorporates ground constraints through the proposed pseudo occupancy method, enhancing the vertical and horizontal constraint effects of ground and planar points. Experimental results on the KITTI dataset demonstrate that the proposed method significantly improves feature extraction and odometry accuracy and efficiency. To enable KNN search using prior maps with limited RAM on UGV, a novel prior map maintenance method is introduced, combining three-layer voxels and the ikd-tree. This method achieves efficient RAM consumption during prior map maintenance. The experimental results confirm its effectiveness in reducing RAM consumption. The proposed method achieves centimeter-level localization accuracy, meeting the requirements for accurate and efficient UGV localization when utilizing prior maps.

In future work, we will focus on the regular updating of the prior map. When there are changes in the scene information, we will incorporate newly acquired point clouds of added objects and remove point clouds of disappeared objects from the previous prior map. The prior map will undergo long-term maintenance to ensure its relevance and accuracy. Furthermore, in order to improve the accuracy of relocation detection, the generation method of the descriptor matrix will be enhanced. The original descriptor matrix only contained the highest point information of the loop key, and will be improved to more comprehensively represent the point cloud information within the loop key, thereby achieving more precise relocation detection and reducing false positives and false negatives.

## Figures and Tables

**Figure 1 sensors-23-06987-f001:**
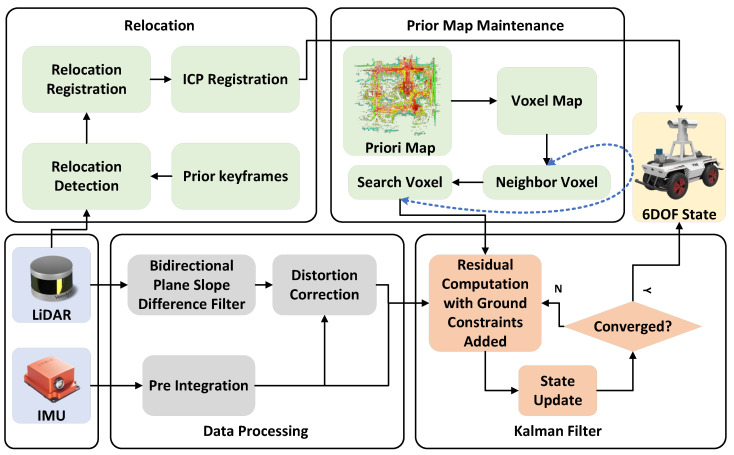
Overall system framework.

**Figure 2 sensors-23-06987-f002:**
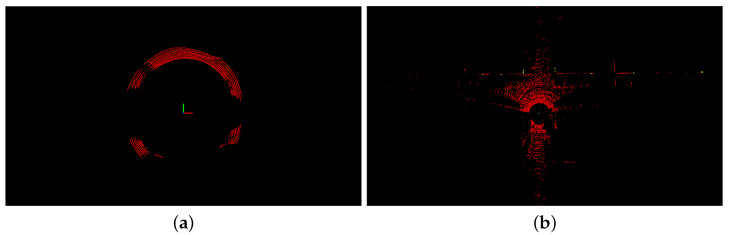
The results of feature extraction. (**a**) is the feature extraction effect of the near-ground end. (**b**) is the feature extraction effect of the far end.

**Figure 3 sensors-23-06987-f003:**
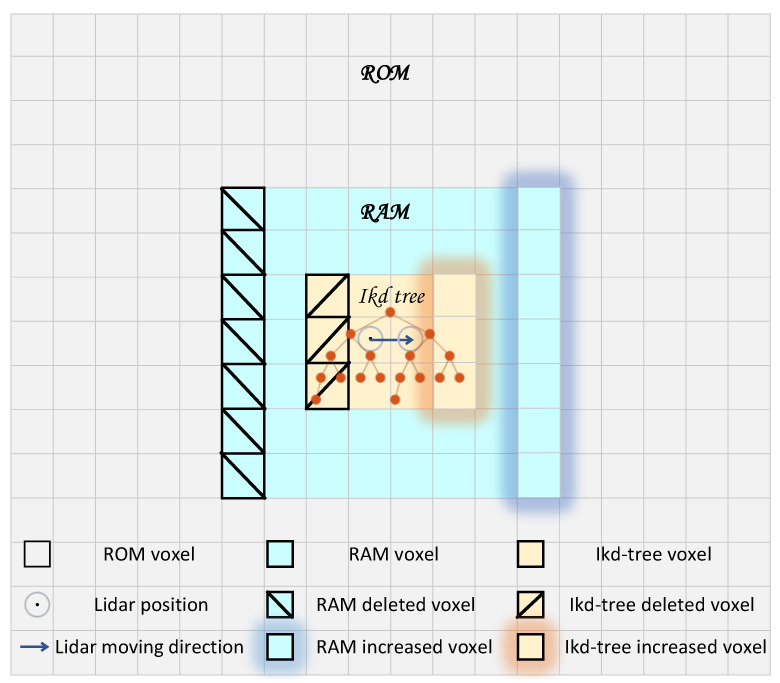
Map maintenance diagram.

**Figure 4 sensors-23-06987-f004:**
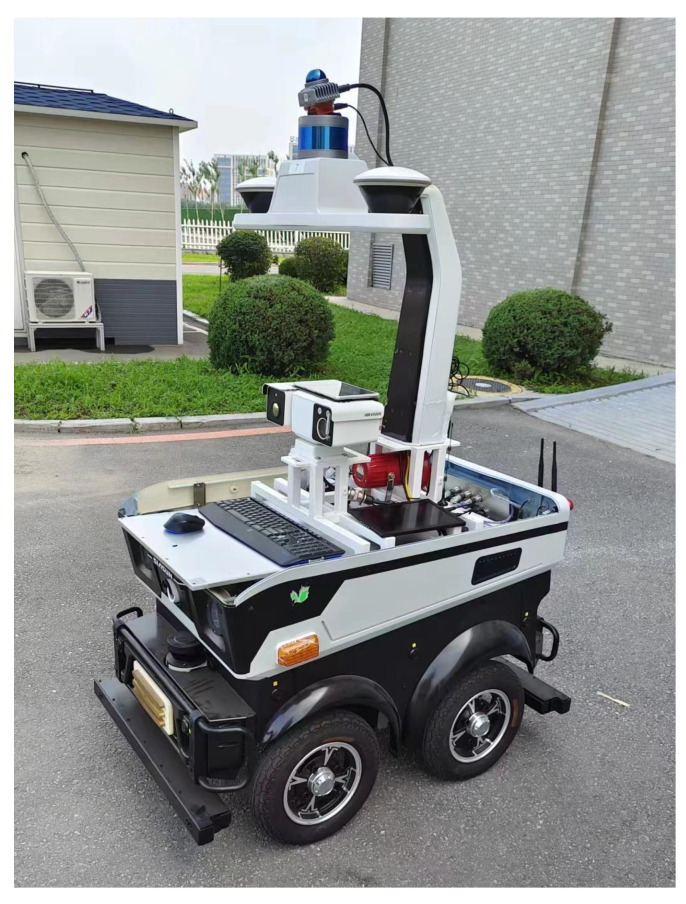
The UGV for collecting indoor and outdoor datasets.

**Figure 5 sensors-23-06987-f005:**
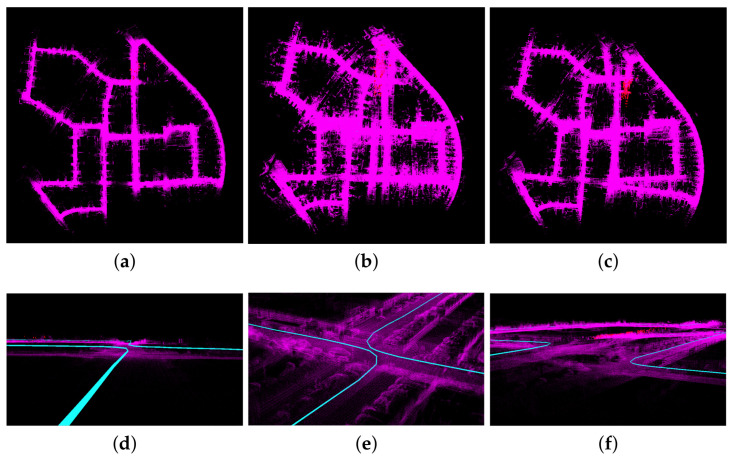
The results of three methods on the KITTI dataset. (**a**) is the result of the combination of proposed odometry method and bidirectional projection plane slope difference filter method. (**b**) is the result of FAST-LIO2. (**c**) is the result of the combination of FAST-LIO2 and the curvature method. (**d**–**f**) are the corresponding local results.

**Figure 6 sensors-23-06987-f006:**
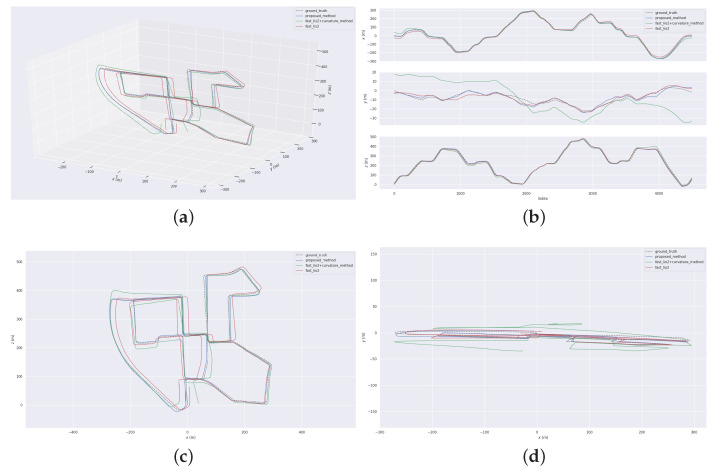
The trajectory map generated by EVO. (**a**) is a 3D trajectory. (**b**) is a three-axis trajectory. (**c**) is a vertical planar trajectory. (**d**) is a horizontal planar trajectory. The black trajectory represents the ground truth trajectory. The blue trajectory corresponds to the trajectory obtained using the proposed method. The green trajectory corresponds to the trajectory obtained using FAST-LIO2 combined with curvature-based feature extraction. The red trajectory corresponds to the trajectory obtained using FAST-LIO2.

**Figure 7 sensors-23-06987-f007:**
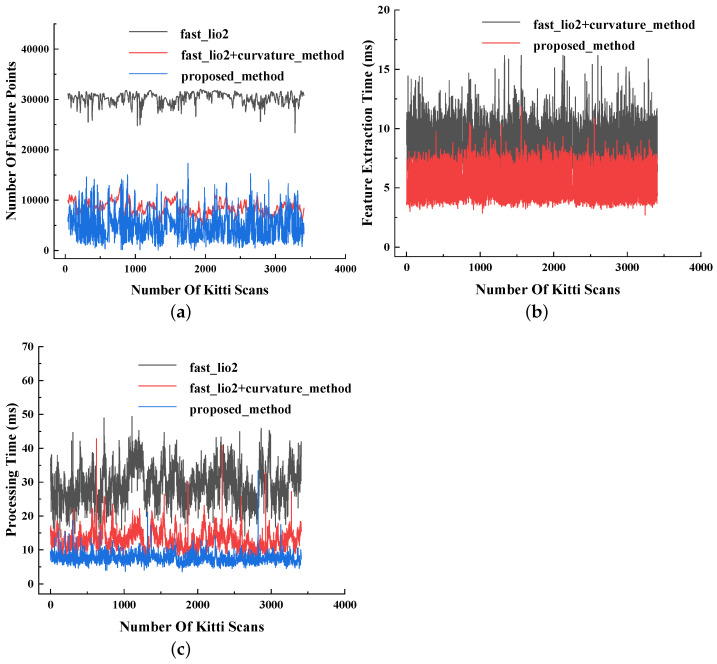
The number of feature points and the curve of time consumption for each part of the KITTI dataset. (**a**) is the curve of the number of feature points of each frame. (**b**) is the curve of the feature extraction time of each frame. (**c**) is the curve of the odometry processing time of each frame.

**Figure 8 sensors-23-06987-f008:**
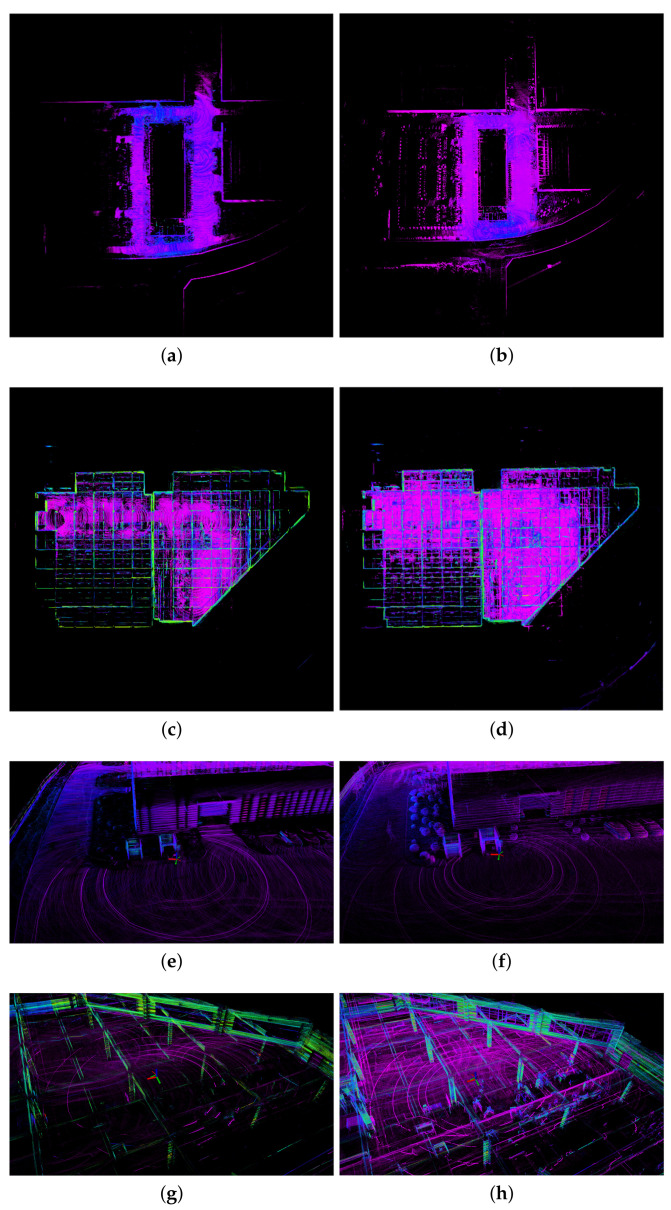
The results of combining different methods with prior maps on indoor and outdoor self-collected datasets. (**a**,**c**,**e**,**g**) are the overall and local figures of the indoor and outdoor results of the proposed odometry method combined with the proposed prior map maintenance method and proposed feature extraction method. (**b**,**d**,**f**,**h**) are the overall and local figures of the indoor and outdoor results of FAST-LIO2 combined with the proposed prior map maintenance method.

**Table 1 sensors-23-06987-t001:** Absolute pose error (APE) of the three methods.

Methods/APE	Max (m)	Mean (m)	Median (m)	Min (m)	Rmse (m)	Std (m)
Proposed method	25.645	11.785	11.232	1.089	12.787	4.968
FAST-LIO2	33.429	16.553	16.991	1.222	18.319	7.847
FAST-LIO2 + curvature	48.196	22.783	22.108	6.580	24.707	9.559

**Table 2 sensors-23-06987-t002:** The mean number of feature points and the time consumption means of each part on the KITTI dataset.

Method	Proposed Method	FAST-LIO2 + Curvature	FAST-LIO2
Number of feature points	5155	8549	30321
Feature extraction time (ms)	5.423	8.483	0
Odometer processing time (ms)	7.553	13.164	29.980
Total time(ms)	12.976	21.647	29.980

**Table 3 sensors-23-06987-t003:** Indoor and outdoor closed-loop error of different methods.

Method/Closed-Loop Error	Δx(m)	Δy(m)	Δz (m)	Total (m)
FAST-LIO2 + Prior map (Outdoor)	0.018	0.092	0.025	0.097
Proposed method + Prior map method (Outdoor)	0.030	0.032	0.0003	0.044
FAST-LIO2 + Prior map (Indoor)	0.018	0.022	0.004	0.029
Proposed method + Prior map method (Indoor)	0.017	0.012	0.0004	0.021

**Table 4 sensors-23-06987-t004:** The time consumption means of each part on the indoor and outdoor datasets.

Method	Scene	FAST-LIO2 + Proposed Prior Map Maintenance Method	Proposed Method + Proposed Prior Map Maintenance Method
Feature extraction time (ms)	Outdoor Indoor	0 0	0.963 1.345
Odometer processing time (ms)	Outdoor Indoor	10.561 12.634	6.426 6.964
Total time (ms)	Outdoor Indoor	10.561 12.634	7.389 8.309

**Table 5 sensors-23-06987-t005:** Test results of map maintenance method.

Method	Average RAM Consumption (GB)	Average RAM Consumption Percentage	KNN Search Average Time (ms)	Number of Points in Ikd-Tree	Map Range in Ikd-Tree (m)	Prior Map Range (m)
Proposed map maintenance method	15	98%	0.00316	2,335,000	240 × 240	1320 × 1320
ikd-tree	5.2	34%	0.00348	12,450,000	1320 × 1320	1320 × 1320

## Data Availability

The data presented in this study is not publicly available at this time but may be obtained from the authors upon reasonable request.

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
