# Peer review of "Robust Localization of Industrial Park UGV and Prior Map Maintenance"

_sensors, 2023, doi:10.3390/s23156987_

Round 1

Reviewer 1 Report

In this manuscript, a tightly-coupled LiDAR-IMU odometry method based on FAST-LIO2 and a maintenance method of prior maps are proposed. The performance of the proposed methods is proved by KITTI dataset, and is useful for localization of UGVs. However, However, several questions should be addressed before it can be accepted.

1.      The abstract provides a brief overview of the study, but it does not include the explanation of the motivation of current work. The reviewer suggests that the authors revise the abstract to provide a clear summary of the research results.

2.   The specific meaning of V_xyin algorithm 2 is not explained.

3.      The specific meaning of u value and the method for calculating u value in line 205 are not explained.

4.      The explanation of R^G_a, R^G_b, R^G_t in line 275 is not accurate and should be changed to the attitude of the IMU at times a, b, and t in the world coordinate system.

5.      The explanation of R_k, R_k+1, R_0 in line 290 is not accurate and should be changed to the attitude of the IMU at times k, k+1, and 0 in the world coordinate system.

6.      The backpropagation formulas (15), (16), and (17) are incorrect.

7.      The methods for detection and correction of reposition in part 4 are not explained clearly.

8.      The values of knn search average time and number of points in ikd-tree in table 5 are not correct, please reivise.

9.      The authors provided some analysis results, the conclusions do not appear to be particularly insightful or thought-provoking. The reviewer suggest that the authors provide a more specific description of the future development direction and practical significance.

Minor editing of English language required.

Reviewer 2 Report

The manuscript “ Robust Localization of Industrial Park UGV and Prior Map Maintenance” addresses a topic of interest to a broad audience and fits the journal's scope. This study presents a tightly-coupled LiDAR-IMU odometry method that leverages prior maps. It also promotes the accuracy of UGV location and reduces RAM usage.

1.     Figure 1. Overall System Framework. Please redraw the order of execution, as it now looks a bit complicated.

2.     Page 14, Line 480  Please describe more about EV and developed by whom.

3.     Figure 6. Could you describe the different colour's meanings and give a colour bar on them?

4.     Figure 7. The legend of this figure is too small to see.

5.     Table 1 What is the unit of this table?

6.     Page 18, Line 574 What is the HTOP tool? Please describe more about the HTOP tool and developed by whom.

7.     Can the conclusion and future work be listed in a way that makes it easier for readers to read?
